# An 11-Year Analysis of Bacterial Foodborne Disease Outbreaks in Zhejiang Province, China

**DOI:** 10.3390/foods11162382

**Published:** 2022-08-09

**Authors:** Lili Chen, Jikai Wang, Ronghua Zhang, Hexiang Zhang, Xiaojuan Qi, Yue He, Jiang Chen

**Affiliations:** Department of Nutrition and Food Safety, Zhejiang Provincial Center for Disease Control and Prevention, 3399 Binsheng Road, Binjiang District, Hangzhou 310051, China

**Keywords:** foodborne disease outbreak, *Vibrio parahaemolyticus*, *Salmonella*

## Abstract

Background: Foodborne diseases are a growing public health problem and contribute significantly to the global burden of disease and mortality. Bacteria are the most common foodborne pathogens. We aimed to explore characteristics of bacterial foodborne disease outbreaks (FBDOs) in Zhejiang Province and to provide data support for foodborne disease prevention and control. Methods: Descriptive statistical methods were used to analyze the data reported by centers for disease control (CDCs) at all levels in Zhejiang Province through Foodborne Disease Outbreaks Surveillance System (FDOSS) during 2010–2020. Results: CDCs in Zhejiang Province reported 517 bacterial FBDOs in 11 years, resulting in 7031 cases, 911 hospitalizations, and 3 deaths. *Vibrio parahaemolyticus* had the highest number of outbreaks, accounting for 58.41% of the total bacterial outbreaks, followed by *Salmonella* (18.38%). In all settings, restaurants (37.14%), staff canteens (11.99%), and households (11.80%) were responsible for the large number of outbreaks. Aquatic products (42.08%), meat and meat products (23.56%), cereals (10.81%), and flour products (9.27%) were the most common single foods reported. Further analysis showed that the settings and food vehicles of outbreaks caused by different pathogens were different. Conclusions: Bacterial outbreaks are the most common type of FBDOs in Zhejiang Province. By analyzing the epidemiological characteristics of common pathogenic bacteria, we can identify the etiology, food, and setting that the government needs to focus on, and issue relevant targeted policies to reduce the number of FBDOs.

## 1. Introduction

Foodborne diseases are a growing public health problem and contribute significantly to the global burden of disease and mortality, causing considerable socioeconomic impact. Every year, nearly one in 10 people around the world fall ill after eating contaminated food, leading to over 420,000 deaths and the loss of 33 million healthy life years (DALYs) [1]. The 2018 World Bank report on the economic burden of the foodborne diseases indicated that the total productivity loss associated with foodborne diseases in low- and middle-income countries was estimated to cost $95.2 billion per year, and the annual cost of treating foodborne illnesses is estimated at $15 billion [2]. During 2003–2017, a total of 19,517 foodborne disease outbreaks (FBDOs) were reported in China, resulting in 235,754 cases, 107,470 hospitalizations, and 1457 deaths, of which bacteria was the main cause, accounting for 34.78% (4657/13,388) of the confirmed outbreaks and causing 99,465 cases (58.52%, 99,465/169,954) [3]. Bacteria are also the main cause of FBDOs in Zhejiang Province [4], one of the most developed coastal provinces in China. To our knowledge, there is no comprehensive analysis of FBDOs caused by bacteria in Zhejiang Province. Analysis of data on FBDOs can be used to identify emerging food safety issues and assess the effectiveness of programs to prevent specific foodborne diseases.

To this end, we analyzed data on bacterial foodborne diseases outbreaks in Zhejiang Province from 2010 to 2020 to obtain basic epidemiological characteristics, including the number and scale of outbreaks, regional distribution, temporal distribution, setting, bacterial species, and food categories, etc. Further etiology-setting and etiology-food combination analysis enables us to obtain the key settings and key foods of a certain kind of bacteria. These results will provide basic data for risk assessment and the determination of regulatory priorities in the future, which will be beneficial to the targeted prevention and control of bacterial foodborne diseases in our province and provide some inspiration for the outbreak prevention and control of bacterial foodborne diseases in other countries or regions.

## 2. Materials and Methods

### 2.1. Outbreak Definition

A foodborne disease outbreak (FBDO) is defined as two or more cases of a similar illness resulting from ingestion of a common food [5]. Diagnostic criteria and principles of management for FBDO of different etiologies were issued by the ministry of health in 1996 and have been used in outbreak investigation ever since [6]. Outbreaks that did not meet these criteria were not reported to the Foodborne Disease Outbreaks Surveillance System (FDOSS).

### 2.2. Data Source

The CDCs at different levels in Zhejiang Province investigated FBDOs within their jurisdiction and reported them to the FDOSS using a uniform format. Information collected by the FDOSS included the location of the outbreak, date of occurrence, setting, etiology, food, number of cases, hospitalizations and deaths caused by the outbreak, and other details.

### 2.3. Statistical Analysis

The data for this study were from outbreak data collected in the FDOSS from 2010 to 2020 and analyzed with Excel 2013. The population data of Zhejiang Province in different years come from the Statistical Yearbook of Zhejiang Province from 2010 to 2020 [7]. We calculated the per capita FBDOs ratio using the average of the total population of Zhejiang Province from 2010 to 2020 as the denominator.

## 3. Results

### 3.1. General Characteristics

A total of 517 bacterial FBDOs were reported to the FDOSS by CDCs in Zhejiang Province from 2010 to 2020, including 7031 cases, 911 hospitalizations, and 3 deaths (Table 1). There were an average of 47 outbreaks and 639.2 cases per year, with 13.6 cases per outbreak. The number of outbreaks was the lowest in 2010 and the highest in 2019; the number of cases was the lowest in 2010 and the highest in 2020. Over the 11-year period, the rate of outbreaks and outbreak-associated cases ranged from 0.32 to 1.47 and 5.37 to 20.71 per 1 million population, respectively. Among the 11 prefecture-level cities, Hangzhou (96 cases, 18.57%) reported the most outbreaks, while Quzhou (21 cases, 4.06%) reported the least outbreaks. Bacterial outbreaks had obvious seasonal characteristics, with 84.91% (439/517) of the outbreaks occurring from May to October (Figure 1). The number of outbreaks has peaked in August each year since 2014.

### 3.2. Etiology

*Vibrio parahaemolyticus* had the highest number of outbreaks, accounting for 58.41% of the total bacterial outbreaks, followed by *Salmonella*, *Staphylococcus aureus*, *Bacillus cereus* and diarrheagenic *Escherichia coli*, accounting for 18.38%, 7.54%, 5.61%, and 5.42%, respectively (Table 2). The largest number of hospitalizations was caused by *Salmonella*, accounting for 43.14% of the total hospitalizations. The three deaths were caused by *Burkholderia gladioli (Pseudomonas cocovenenans* subsp. *farinofermentans*)(one death), *Vibrio parahaemolyticus* (one death) and *Bacillus cereus* (one death).

### 3.3. Setting

In all settings, restaurants (192 outbreaks, 37.14%), staff canteens (62 outbreaks, 11.99%) and households (61 outbreaks, 11.80%) were responsible for the large number of outbreaks (Table 3). The largest number of cases occurred in restaurants (2374 cases, 33.76%), followed by the school canteen (1401 cases, 19.93%) and the staff canteen (949 cases, 13.50%). Outbreaks in restaurants (281 hospitalizations, 30.85%), bakeries (166 hospitalizations, 18.22%), school canteens (123 hospitalizations, 13.50%), and staff canteens (112 hospitalizations, 12.29%) resulted in relatively high numbers of hospitalizations. Two of the three deaths occurred in households and one in the restaurant.

### 3.4. Food

Confirmed foods were reported for 340 (65.76%, 340/517) outbreaks and 259 (50.10%, 259/517) outbreaks were attributed to a single food (Table 4). Aquatic products (42.08%, 109/259), meat and meat products (23.55%, 61/259), cereals (10.81%, 28/259) and flour products (9.27%, 24/259) were common single food reported. For single food, aquatic products accounted for the largest number of cases (1073 cases, 15.26%), followed by meat and meat products (842 cases, 11.98%), flour products (732 cases, 10.41%), and cereals (310 cases, 4.41%). The single food that caused the most hospitalizations was flour products (261 hospitalizations, 28.65%), followed by meat and meat products (105 hospitalizations, 11.53%), aquatic products (77 hospitalizations, 8.45%), and cereals (73 hospitalizations, 8.01%). Two deaths were caused by black fungus and fried rice, respectively, while the other death was caused by unknown food.

### 3.5. Region and Etiology

The proportion of *Vibrio parahaemolyticus* ranked first in 11 prefecture-level cities, among which Ningbo (79.07%, 34/43) has the highest proportion, followed by Zhoushan (78.95%, 30/38) (Figure 2). *Salmonella* was the second most common pathogen in most regions except Hangzhou and Huzhou. The proportion of *Staphylococcus aureus* was higher in Wenzhou (18.67, 14/75), Lishui (13.89, 5/36) and Hangzhou (11.46%, 11/96), and the proportion of *Bacillus cereus* was higher in Jinhua (20.75%, 11/53). In addition, the proportion of diarrheagenic *Escherichia coli* was relatively high in Huzhou (22.22%, 6/27) and Hangzhou (13.54%, 13/96).

### 3.6. Setting and Etiology

The settings of outbreaks caused by various pathogens were different (Figure 3). *Vibrio parahaemolyticus* outbreaks mainly occurred in restaurants (51.99%, 157/302), staff canteens (13.58%, 41/302) and rural banquets (13.25%, 40/302); *Salmonella* outbreaks mainly occurred in households (26.32%, 29/95), restaurants (15.79%, 15/95), and bakeries (10.53%, 10/95). For households, *Salmonella* (47.54%, 29/61) and *Vibrio parahaemolyticus* (36.07%, 22/61) both accounted for a high proportion of outbreaks. In addition, the main pathogens that occurred in fast food restaurants, bakeries and street stalls were *Vibrio parahaemolyticus* (71.43%, 10/14), *Salmonella* (90.91%, 10/11) and *Bacillus cereus* (40%, 6/15), respectively. Diarrheagenic *Escherichia coli* (29.17%, 14/48) and *Staphylococcus aureus* (25%, 12/48) were the main pathogens for outbreaks occurred in school canteens.

### 3.7. Food and Etiology

The analysis of the main pathogenic bacteria and single foods showed that outbreaks caused by different pathogens tended to occur in different foods (Table 5). *Vibrio parahaemolyticus* outbreaks were mainly caused by aquatic products (72.18%, 96/133) and meat and meat products (18.05%, 24/133); aquatic products mainly included crustaceans (42.20%, 46/109), mollusks (32.11%, 35/109) and fish (19.27%, 21/109); meat and meat products mainly included ready-to-eat cooked meat cold dishes. The food that caused *Salmonella* outbreaks were mainly flour products (33.33%, 18/54) and meat and meat products (29.63%, 16/54). The flour products were mainly sandwiches, and the meat and meat products were mainly ready-to-eat cold cooked meat. *Bacillus cereus* outbreaks were mainly caused by cereals (84.62%, 22/26), mainly cooked rice. The foods responsible for the diarrheagenic *Escherichia coli* outbreaks were mainly caused by meat and meat products (66.67%, 6/9).

## 4. Discussion

The number of bacterial outbreaks increased significantly between 2010 and 2016 and has remained largely stable since 2017. This may be due to the gradual improvement of the outbreak surveillance system (upgrading of the system operating environment and optimization of the reporting process) and the increased awareness of reporting at all levels of CDCs and hospitals. *Vibrio parahaemolyticus* (58.41%) was the most common pathogen in Zhejiang Province, followed by *Salmonella* (18.38%), which are consistent with the studies in mainland China [8,9], but different from some provinces [10,11] due to geographical environment and dietary habits. The highest number of bacterial FBDOs occurred in restaurants, mainly due to cross contamination, improper storage or inadequate cooking, which has been illustrated by some domestic studies [12,13,14]. Bacterial outbreaks mostly occurred in summer and autumn (May to October), which is consistent with data reported in mainland China from 2003 to 2017 [3]. The average daily maximum temperature in Zhejiang Province from May to October is 24 to 35 °C, and the average daily minimum temperature is 16 to 27 °C. Warmer weather speeds up bacterial reproduction, increasing the likelihood of bacterial outbreaks. A study showed that bacterial pathogens of gastrointestinal infection were positively correlated with ambient temperature, and the higher the temperature, the faster the replication [15]. Therefore, attention should be paid to the prevention and control of bacterial outbreaks in summer and autumn, especially in August.

There are regional differences in the distribution of foodborne pathogens. The top four pathogenic bacteria species in Zhejiang Province were *Vibrio parahaemolyticus*, *Salmonella*, *Staphylococcus aureus* and *Bacillus cereus*, the same as the monitoring data from 2003 to 2017 in Mainland China [3], but different from the common bacteria species in the United States [16], Brazil [17] and the European Union [18]. In addition, although the main pathogenic bacteria species are roughly the same across provinces in China, the rank is different. *Vibrio parahaemolyticus* has caused the most outbreaks in coastal provinces and *Salmonella* has caused the largest proportion of outbreaks in inland provinces [19]. Similarly, in the 11 prefection-level cities of Zhejiang Province, the proportion of different pathogenic bacteria is also different. Although the proportion of *Vibrio parahaemolyticus* in each prefection-level city was the highest, the proportion in coastal cities Ningbo and Zhoushan was 79.07% and 78.95%, respectively, while the proportion in inland cities Jinhua was only 39.62%, which was mainly related to dietary habits and food processing methods. On the other hand, with the rapid development of logistics and the change of people’s dietary habits, the proportion of *Vibrio parahaemolyticus* in some inland cities is not low. For example, the proportion of *Vibrio parahaemolyticus* in Shaoxing reaches 74.19%. Therefore, different regions should take targeted measures against the main pathogens in their locality.

*Vibrio parahaemolyticus* is considered to be the most prevalent food-poisoning bacterium associated with seafood consumption [20]. Our study showed that *Vibrio parahaemolyticus* was also the leading cause of FBDOs and outbreak-associated cases in Zhejiang Province, mainly involving aquatic products, meat and meat products. Outbreaks associated with aquatic products were mainly associated with eating raw/undercooked seafood and another study found that aquatic products in Zhejiang Province had a high detection rate of *V**ibrio parahaemolyticus* [21]. Outbreaks caused by meat and meat products (mainly ready-to-eat cooked meat) were associated with cross-contamination. Although *Vibrio parahaemolyticus* was mainly derived from animal seafood, it is also worth paying attention to the outbreaks of *Vibrio parahaemolyticus* caused by ready-to-eat cooked meat due to cross-contamination in food preparation. In addition, given that *Vibrio parahaemolyticus* outbreaks mainly occurred in restaurants (51.99%, 157/302), supervision of restaurants and training of chefs and key management personnel should be strengthened. Food Handlers should adhere to the Five Keys to Safer Food manual [22] to reduce the occurrence of FBDOs.

*Salmonella* is the second leading pathogen of bacterial FBDOs in Zhejiang Province, accounting for 18.38% of bacterial foodborne disease, although much lower than the 22.5% of all outbreaks reported by the European Union [18], but caused 43.14% of hospitalizations cases, resulting in a huge disease burden. *Salmonellosis* is linked to the consumption of *Salmonella*-contaminated food products mostly from poultry, pork, and egg products [23]. Generally speaking, food vehicles of *Salmonella* outbreaks vary in different countries or regions due to food processing methods and dietary habits [24,25,26,27,28]. However, with the advance of globalization, in addition to traditional domestic food vehicles, food from abroad will also become new food vehicles for outbreaks. For example, sandwiches in Zhejiang Province have caused more and more outbreaks in recent years. Our study found that the main food vehicle of *Salmonella* outbreaks in Zhejiang Province was ready-to-eat food, including cold processed cakes such as sandwiches and ready-to-eat cold cooked meat. *Salmonella* outbreaks caused by sandwiches and other cold processed cakes have been reported in other provinces in China [29,30,31]. The main causes of such outbreaks are raw and cooked cross-contamination of worktables in sandwich factories, improper handling of eggs, inadequate hand hygiene facilities in some factories, exposure to improper holding temperature during delivery, and improper storage in retail stores. Effective measures should be taken to prevent such outbreaks. For example, food handlers should observe proper hand hygiene and handle raw and cooked food separately to avoid cross-contamination when preparing food. Sandwiches are transported in the cold chain all the way to the retail store and refrigerated until they are sold at the retail store.

In our study, *Bacillus cereus* caused one death and the food that caused the outbreak was Chinese-style fried rice. Although rare, deaths from foodborne diseases caused by *Bacillus cereus* have also been reported [32,33]. *Bacillus cereus* outbreaks in Zhejiang province were mainly caused by cereals (leftover rice), which can be eaten directly after heating or made into Chinese-style fried rice. A study from Sri Lanka showed a 56% detection rate of *Bacillus cereus* in 200 samples of Chinese-style fried rice [34]. Another study in Zhejiang Province showed that the detection rate of *Bacillus cereus* in boiled rice reached 71.4% [35]. Leftover food contaminated with *Bacillus cereus*, if not properly preserved before consumption, can lead to reproduction and toxin production, which can ultimately lead to foodborne diseases.

In addition, two outbreaks caused by the consumption of long-term soaked black fungus were reported in Zhejiang Province in July 2018 and July 2019, respectively. The pathogen responsible for both outbreaks was *Burkholderia gladioli* (*Pseudomonas cocovenenans* subsp. *farinofermentans*), which is widely found in nature. The most suitable temperature for growth of *Burkholderia gladioli* (*Pseudomonas cocovenenans* subsp. *farinofermentans*) is 37 °C, and the temperature for toxin production is 26 °C [36]. The average temperature in July in Zhejiang province was 27 °C~35 °C, which was favorable for the propagation and toxicity of the pathogen. Black fungus is a traditional Chinese food. Investigations showed that black fungus (dried) itself was not poisonous in both outbreaks. The causes of two outbreaks were that the black fungus was contaminated by *Burkholderia gladioli* (*Pseudomonas cocovenenans* subsp. *farinofermentans*) in the process of soaking for a long time, and it multiplied and produced toxins at the appropriate temperature. In order to avoid such outbreaks, it is recommended that people do not eat black fungus soaked for too long, especially in summer and autumn.

There are some limitations in this study. Although our surveillance system has greatly improved the outbreak reporting rate, the epidemiological investigation cannot be carried out smoothly due to the patient’s lack of cooperation (amongst other reasons), and there will be some missing report rate eventually. For some reason, the investigation of some outbreaks was incomplete, and the etiology and food were not known. Our study draws more conclusions from confirmed outbreaks, but conclusions from confirmed outbreaks may not apply to all outbreaks. In addition, diarrheagenic *Escherichia coli* was not further classified. We will focus on this topic in the upcoming outbreak surveillance.

## 5. Conclusions

Proper food preparation can prevent most foodborne diseases. To reduce the outbreaks of bacterial foodborne diseases, food handlers should adhere to the Five Keys to Safer Food manual. In addition, the government should adopt targeted prevention and control strategies according to the characteristics of different pathogenic bacteria (high-incidence season, high-incidence area, high-incidence setting, high-incidence food, etc.). In the future, CDCs will further strengthen FBDOs surveillance and data analysis capabilities to support government efforts to reduce the incidence of foodborne disease.

## Figures and Tables

**Figure 1 foods-11-02382-f001:**
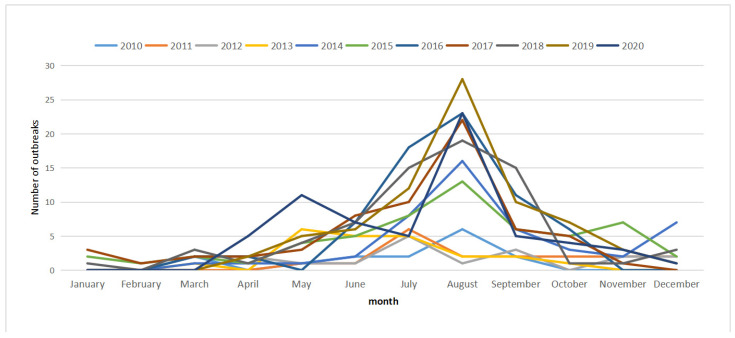
Year-month distribution of outbreaks of bacterial FBDOs in Zhejiang Province, 2010–2020.

**Figure 2 foods-11-02382-f002:**
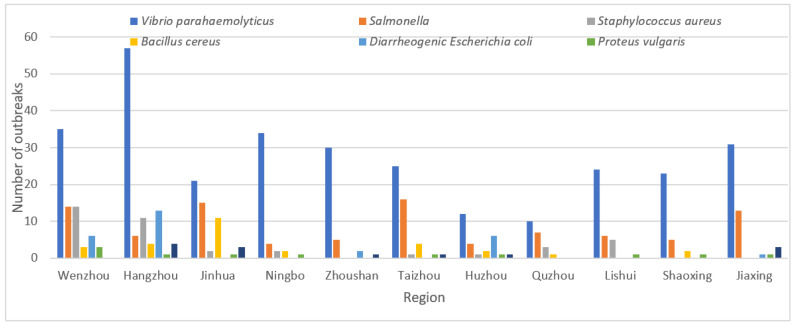
Region-etiology distribution of outbreaks of bacterial FBDOs in Zhejiang Province, 2010–2020.

**Figure 3 foods-11-02382-f003:**
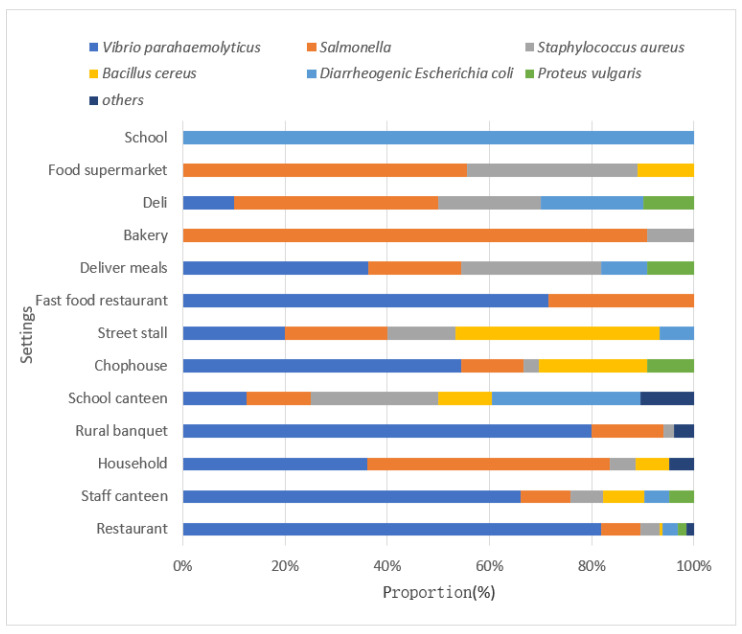
Setting-etiology distribution of outbreaks of bacterial FBDOs in Zhejiang Province, 2010–2020.

**Table 1 foods-11-02382-t001:** Annual distribution of outbreaks of bacterial FBDOs in Zhejiang Province, 2010–2020.

Year	Outbreaks		Cases		Hospitalizations		Deaths	
	*n*	%	*n*	%	*n*	%	*n*	%
2010	15	2.90	255	3.63	3	0.33	0	0.00
2011	18	3.48	428	6.09	160	17.56	0	0.00
2012	19	3.68	277	3.94	47	5.16	0	0.00
2013	22	4.26	654	9.30	133	14.60	0	0.00
2014	47	9.09	467	6.64	42	4.61	0	0.00
2015	56	10.83	583	8.29	44	4.83	1	33.33
2016	69	13.35	863	12.27	181	19.87	1	33.33
2017	63	12.19	671	9.54	46	5.05	0	0.00
2018	70	13.54	926	13.17	128	14.05	0	0.00
2019	74	14.31	857	12.19	59	6.48	1	33.33
2020	64	12.38	1050	14.93	68	7.46	0	0.00
Total	517	100.00	7031	100.00	911	100.00	3	100.00

**Table 2 foods-11-02382-t002:** Etiological distribution of bacterial FBDOs in Zhejiang Province from 2010 to 2020.

Etiology	Outbreaks		Illnesses		Hospitalizations		Deaths	
	Number	%	Number	%	Number	%	Number	%
*Vibrio parahaemolyticus*	302	58.41	3469	49.34	311	34.14	1	33.33
*Salmonella*	95	18.38	1559	22.17	393	43.14	0	0.00
*Staphylococcus aureus*	39	7.54	512	7.28	131	14.38	0	0.00
*Bacillus cereus*	29	5.61	307	4.37	38	4.17	0	0.00
Diarrheagenic *Escherichia coli* ^a^	28	5.42	784	11.15	10	1.10	0	0.00
*Proteus vulgaris* ^b^	11	2.13	77	1.10	18	1.98	0	0.00
*Aeromonas* ^b^	4	0.77	127	1.81	6	0.66	0	0.00
*Burkholderia gladioli* (*Pseudomonas cocovenenans* subsp. *farinofermentans)*	2	0.39	4	0.06	2	0.22	2	66.67
*Campylobacter jejuni*	2	0.39	132	1.88	1	0.11	0	0.00
others	5	0.97	60	0.85	1	0.11	0	0.00
Total	517	100.00	7031	100.00	911	100.00	3	100.00

^a^ In China’s national food safety standard GB4789.6-2016, diarrheagenic *Escherichia coli* was defined as a kind of *Escherichia coli* that can cause diarrhea symptoms in humans and can cause human illness through contaminated food. Diarrheagenic *Escherichia coli* mainly includes EPEC, ETEC, EAEC, EIEC, etc. ^b^ These microorganisms are not typically pathogenic but common present in food samples which need to be flagged for further inquiry and treated with caution.

**Table 3 foods-11-02382-t003:** Setting distribution of outbreaks of bacterial FBDOs in Zhejiang Province from 2010 to 2020.

Setting	Outbreaks		Cases		Hospitalizations		Deaths	
	*n*	%	*n*	%	*n*	%	*n*	%
Restaurant	192	37.14	2374	33.76	281	30.85	1	33.33
Staff canteen	62	11.99	949	13.50	112	12.29	0	0.00
Household	61	11.80	277	3.94	40	4.39	2	66.67
Rural banquet	50	9.67	652	9.27	65	7.14	0	0.00
School canteen	48	9.28	1401	19.93	123	13.50	0	0.00
Chophouse	33	6.38	225	3.20	24	2.63	0	0.00
Street stall	15	2.90	148	2.10	0	0.00	0	0.00
Fast food restaurant	14	11.99	175	2.49	12	1.32	0	0.00
Deliver meals	11	2.13	106	1.51	46	5.05	0	0.00
Bakery	11	2.13	466	6.63	166	18.22	0	0.00
Deli	10	1.93	80	1.14	15	1.65	0	0.00
Food supermarket	9	1.74	150	2.13	23	2.52	0	0.00
School	1	0.19	28	0.40	4	0.44	0	0.00
Total	517	100	7031	100.00	911	100.00	3	100.00

**Table 4 foods-11-02382-t004:** Food distribution of outbreaks of bacterial FBDOs in Zhejiang Province, 2010–2020.

Food	Outbreaks		Cases		Hospitalizations		Deaths	
	Number	%	Number	%	Number	%	Number	%
Aquatic products	109	21.08	1073	15.26	77	8.45	0	0.00
Meat and meat products	61	11.80	842	11.98	105	11.53	0	0.00
Cereals	28	5.42	310	4.41	73	8.01	0	0.00
Flour products	24	4.64	732	10.41	261	28.65	0	0.00
Vegetable	10	1.93	154	2.19	28	3.07	0	0.00
Egg and egg products	9	1.74	73	1.04	15	1.65	0	0.00
Takeaway food	6	1.16	84	1.19	39	4.28	0	0.00
Bean products	5	0.97	86	1.22	0	0.00	0	0.00
Fungus (black fungus)	2	0.39	4	0.06	2	0.22	2	66.67
Other single foods	5	0.97	53	0.75	4	0.44	0	0.00
Mixed dishes	81	15.67	1223	17.39	156	17.12	0	0.00
Unknown	177	34.24	2397	34.09	151	16.58	1	33.33
Total	517	100.00	7031	100.00	911	100.00	3	100.00

**Table 5 foods-11-02382-t005:** Food-etiology distribution of bacterial FBDOs in Zhejiang Province, 2010–2020.

Bacteria	Food										Total
	Aquatic Products	Meat and Meat Products	Cereals	Flour Products	Vegetable	Egg and Egg Products	Takeaway Food	Bean Products	Fungus (Black Fungus)	Other Single Foods	
*Vibrio parahaemolyticus*	96	24	0	0	1	3	7	1	0	1	133
*Salmonella*	7	16	0	18	1	1	2	6	0	3	54
*Staphylococcus aureus*	3	9	6	3	4	0	0	1	0	0	26
*Bacillus cereus*	1	0	22	2	0	0	0	1	0	0	26
Diarrheagenic *Escherichia coli*	0	6	0	0	0	1	1	0	0	1	9
*Proteus vulgaris*	1	3	0	1	0	0	0	0	0	0	5
Other	1 ^a^	3 ^b^	0	0	0	0	0	0	2 ^c^	0	6
Total	109	61	28	24	6	5	10	9	2	5	259

^a^ Aeromonas (one outbreak). ^b^ Clostridium perfringens (one outbreak); Campylobacter jejuni (two outbreaks). ^c^ Burkholderia gladioli (Pseudomonas cocovenenans subsp. farinofermentans) (two outbreaks).

## Data Availability

The study of the data can be downloaded from the Foodborne Disease Outbreaks Surveillance System (https://sppt.cfsa.net.cn/goto (accessed on 1 June 2021)), and the data is not currently open.

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
