# Peer review of "An 11-Year Analysis of Bacterial Foodborne Disease Outbreaks in Zhejiang Province, China"

_foods, 2022, doi:10.3390/foods11162382_

Round 1

Reviewer 1 Report

This manuscript gives useful information on the outbreaks in Zhejiang Province, but I am not convinced of the certainty of the data since some of the pathogens mentioned are not standard names and the results are different from other outbreak studies. How well was the lab analysis carried out? What is diarrheagenic E. coli? Burkholderia gladioli may be the correct name for Pseumonas cocovenenans. Deaths from Bacillus cereus are very rare. Did the epidemiology identify specific foods? A black fungus is mentioned, give more detail on that. I agree that many outbreak investigations are incomplete, and it is not always possible to draw conclusions from them. At least, a limitation clause should be added. I also, think it would be useful for the reader to have two or three examples of specific investigations.

Line 46. Change “We will analyze data on..” to “To this end, we analyzed data…”

Line 98. diarrheagenic [no italics] Escherichia coli,[italics]. What is this organism, probably a combination of different species, e.g., ETEC, EPEC, EHEC, EIEC, etc.?

Lines 100-102. The three deaths occurred during this time period: one each from were caused by Pseudomonas cocovenenans [italics] subsp [no italics]  farinofermantans [italics], Vibrio parahaemolyticus, and Bacillus cereus.

Table 3. I am not sure what Proteusbacillus vulgaris is. Maybe just Proteus vulgaris, although it is not a known pathogen. Also, Campylobacter.

Table 4. This needs attention to detail; under year you have establishments and some are first letter upper case and others not.

Text and tables. Aquatic products needs more specifics. If V. parahaemolyticus is the pathogen, most foods would be different types of shellfish. Some pathogens prefer fish over shellfish, but we do not see that detail in this manuscript..

Lines 165-167. This sentence needs rewriting. The foods responsible for the Salmonella outbreaks were mainly flour products (33.33%, 18/54), mainly sandwiches and meat and meat products (29.63%, 16/54), mainly ready-to-eat cold cooked meat. Also, I am surprised how many flour products were linked to Salmonella. This is not the case in most other foodborne outbreak surveys.

Line 46. Change “We will analyze data on..” to “To this end, we analyzed data…”

Line 98. diarrheagenic [no italics] Escherichia coli,[italics]. What is this organism, probably a combination of different species, e.g., ETEC, EPEC, EHEC, EIEC, etc.?

Lines 100-102. The three deaths occurred during this time period: one each from were caused by Pseudomonas cocovenenans [italics] subsp [no italics]  farinofermantans [italics], Vibrio parahaemolyticus, and Bacillus cereus.

Table 3. I am not sure what Proteusbacillus vulgaris is. Maybe just Proteus vulgaris, although it is not a known pathogen. Also, Campylobacter.

Table 4. This needs attention to detail; under year you have establishments and some are first letter upper case and others not.

Text and tables. Aquatic products needs more specifics. If V. parahaemolyticus is the pathogen, most foods would be different types of shellfish. Some pathogens prefer fish over shellfish, but we do not see that detail in this manuscript..

Lines 165-167. This sentence needs rewriting. The foods responsible for the Salmonella outbreaks were mainly flour products (33.33%, 18/54), mainly sandwiches and meat and meat products (29.63%, 16/54), mainly ready-to-eat cold cooked meat. Also, I am surprised how many flour products were linked to Salmonella. This is not the case in most other foodborne outbreak surveys.

Author Response

Response: Thanks for your wonderful comments. Under your careful instructions, we have considered all of your precious suggestions in the revised manuscript and we have responded the questions you proposed point by point as follow.  

  1. How well was the lab analysis carried out? What is diarrheagenic E. coli? Burkholderia gladioli may be the correct name for Pseumonas cocovenenans.

Line 98. diarrheagenic [no italics] Escherichia coli,[italics]. What is this organism, probably a combination of different species, e.g., ETEC, EPEC, EHEC, EIEC, etc.?

Response: We all appreciate your valuable comments. Our laboratory testing of each kind of pathogenic bacteria is in strict accordance with the national standard testing methods. In China's national food safety standard GB4789.6-2016, diarrheagenic Escherichia coli was defined as a kind of Escherichia coli that can cause diarrhea symptoms in humans and can cause human illness through contaminated food. Diarrheagenic Escherichia coli mainly includes EPEC, ETEC, EAEC, EIEC, etc. We have also checked GB 4789.29-2020. Pseudomonas cocovenenans subsp. farinofermentans is accurately spelled Burkholderia gladioli (Pseudomonas cocovenenans subsp. farinofermentans).

  1. Deaths from Bacillus cereusare very rare. Did the epidemiology identify specific foods?

Response: Indeed, as you said, deaths caused by Bacillus cereus are very rare, and we have only found one death in 11 years of surveillance. Although being rare, fatal cases of B. cereus-induced food poisoning have been

reported (https://journals.asm.org/doi/10.1128/JCM.05129-11

; https://doi.org/10.1542/peds.2009-2319). The food in this outbreak is  Chinese-style fried rice. Chinese are very fond of fried rice, which is cooked with leftover food. If the leftover food is not kept properly, it is easy to be contaminated with Bacillus cereus, which has been found in our surveillance for many years. A study from Sri Lanka also showed a 56% detection rate of Bacillus cereus in 200 samples of Chinese-style fried rice ( DOI: 10.1089/fpd.2011.0969). We have added a discussion on this topic in the revised manuscript.

  1. A black fungusis mentioned, give more detail on that.

Response: We have provided more details about two outbreaks caused by black fungus. Please check the revised manuscript.

  1. At least, a limitation clause should be added.

Response: Thanks for your good suggestion. Some limitations of this study need to be explained. Please check the revised manuscript.

  1. I also, think it would be useful for the reader to have two or three examples of specific investigations.

Response: Thank you for your suggestion. We understand your point of view, but the purpose of this article is to describe the epidemiological characteristics of bacterial foodborne disease outbreaks in general and provide data support for the prevention and control of foodborne diseases. It is therefore inappropriate to give an epidemiological investigation of some outbreak. If Foods is interested in this, we can write another article on this topic. Many of the investigations are well worth writing about.

  1. Line 46. Change “We will analyze data on..” to “To this end, we analyzed data…”

Response: We have made the modification according to your suggestion. Please check the revised manuscript.

  1. Lines 100-102. The three deaths occurred during this time period: one each from were caused by Pseudomonas cocovenenans [italics] subsp [no italics]  farinofermantans [italics], Vibrio parahaemolyticus, and Bacillus cereus.

Response: We have made the modification according to your suggestion. Please check the revised manuscript.

  1. Table 3. I am not sure what Proteusbacillus vulgaris is. Maybe just Proteus vulgaris, although it is not a known pathogen. Also, Campylobacter.

Response: Proteus vulgaris; Campylobacter jejuni.

  1. Table 4. This needs attention to detail; under year you have establishments and some are first letter upper case and others not.

Response: We have made the modification according to your suggestion. Please check the revised manuscript.

  1. Text and tables. Aquatic productsneeds more specifics. If V. parahaemolyticus is the pathogen, most foods would be different types of shellfish. Some pathogens prefer fish over shellfish, but we do not see that detail in this manuscript.

Response: We have given the specific classification of aquatic products. Aquatic products mainly included crustaceans (42.20%, 46/109), mollusks (32.11%, 35/109) and fish (19.27%, 21/109), etc.

  1. Lines 165-167. This sentence needs rewriting. The foods responsible for the Salmonella outbreaks were mainly flour products (33.33%, 18/54), mainly sandwiches and meat and meat products (29.63%, 16/54), mainly ready-to-eat cold cooked meat. Also, I am surprised how many flour products were linked to Salmonella. This is not the case in most other foodborne outbreak surveys.

Response: We have rewritten the sentence. The flour products responsible for Salmonella outbreaks were mainly cold processed cakes such as sandwiches. Sandwiches are not a traditional food in our province and therefore were not the main food responsible for the Salmonella outbreaks in the early stages of surveillance. But with globalization, sandwiches have become a popular food among Chinese people. Salmonella outbreaks caused by sandwiches have also come to our attention. We also addressed this in the Discussion section. Please check the revised manuscript.

Reviewer 2 Report

1. The question is  well-defined but not very original. The authors described a similar topic in the article from 2022 (No. 4 in the References:

Chen, L., Sun, L., Zhang, R. et al. Surveillance for foodborne disease outbreaks in Zhejiang Province, China, 2015–2020. BMC 286 Public Health 22, 135 (2022). https://doi.org/10.1186/s12889-022-12568-4 ).

2. The work fits the journal scope.

3. The results interpreted appropriately. The study would provide a better picture of the epidemiological situation, extended to other regions. Conclusions justified and supported by the results.

4. I propose to replace table 2, 6, 7 with a graphs with trend lines. The text requires careful analysis, e.g. in lines 33,36,41,200 there is no space before [].

5. The study correctly designed and technically sound. The data robust enough to draw conclusions.

6. The conclusions are interesting for the readership of the journal, but the paper can be of interest to a limited number of people due to the describe data from one region.

Author Response

Response: Thanks for your wonderful comments. Under your careful instructions, we have considered all of your precious suggestions in the revised manuscript and we have responded the questions you proposed point by point as follow.  

  1. The question is  well-defined but not very original. The authors described a similar topic in the article from 2022 (No. 4 in the References:

Chen, L., Sun, L., Zhang, R. et al. Surveillance for foodborne disease outbreaks in Zhejiang Province, China, 2015–2020. BMC 286 Public Health 22, 135 (2022). https://doi.org/10.1186/s12889-022-12568-4 ).

Response: Thanks for your comments. The above article is an epidemiological analysis of all foodborne disease outbreaks reported in Zhejiang Province from 2015 to 2020. It describes all etiologies, including pathogenic bacteria, viruses, poisonous mushrooms, poisonous plants and their toxins, poisonous animals and their toxins, chemicals, etc. Through the above analysis, we found that pathogenic bacteria were the cause of the most outbreaks, so this manuscript provides a detailed analysis of this main cause over a longer period (2010-2020). Although pathogenic bacteria have been mentioned in previous articles, they are not as detailed as our current study.

  1. The work fits the journal scope.

Response: Thanks for your comments.

  1. The results interpreted appropriately. The study would provide a better picture of the epidemiological situation, extended to other regions. Conclusions justified and supported by the results.

Response: Thanks for your comments.

  1. I propose to replace table 2, 6, 7 with a graphs with trend lines. The text requires careful analysis, e.g. in lines 33,36,41,200 there is no space before [].

Response: Thanks for your kind suggestion. We have replaced Table 2, 6 ,7 with Figure 1, 2 ,3. In addition, we carefully checked the text and added the space before [].

  1. The study correctly designed and technically sound. The data robust enough to draw conclusions.

Response: Thanks for your comments.

  1. The conclusions are interesting for the readership of the journal, but the paper can be of interest to a limited number of people due to the describe data from one region.

Response: Thanks for your comments. Although this article is based on data from one region, foodborne disease outbreaks are a public health problem faced by the whole world. Other countries or regions can learn about the epidemiological characteristics of foodborne disease outbreaks in our region and get some inspiration, so as to better carry out the prevention and control work of foodborne disease in their regions.

Reviewer 3 Report

The manuscript entitled “Surveillance for bacterial foodborne disease outbreaks in Zhejiang Province, China, 2010-2020” aims to explore the characteristics of bacterial foodborne disease outbreaks (FBDOs) in a China province and provide data support for foodborne disease prevention and control. The applied methodology is the classic epidemiological data collection.

Regarding the manuscript, I would like to comment on the following:

The presentation of the data is very descriptive and sometimes repetitive in the discussion section. From the epidemiological standpoint, it would also be interesting to include the population of Zhejiang Province and a better description of the age group affected by the different foodborne diseases. Furthermore, it would be interesting to determine the incidence per 100,000 inhabitants within this surveillance.

Table 2 could be shown as a figure.

Some information in the text could be summarized, reducing information already shown in the tables. Why is the data not compared with those of the European Union? European Unions have been reporting the incidence of foodborne illness for many years. See: https://www.ecdc.europa.eu/en/all-topics-z/food-and-waterborne-diseases-and-zoonoses/surveillance-and-disease-data/eu-one-health. Database: https://www.efsa.europa.eu/en/microstrategy/FBO-dashboard

Many data with similar results appear in an article published by the same authors, see “Chen, L., Sun, L., Zhang, R., Liao, N., Qi, X., & Chen, J. (2022). Surveillance for foodborne disease outbreaks in Zhejiang Province, China, 2015–2020. BMC Public Health, 22(1), 2015–2020. https://doi.org/10.1186/S12889-022-12568-4”. Although the time is indeed limited, and it includes other non-biological hazards. However, part of the conclusions, as expected, are very similar.

Line 97: Salmonella enterica is non-typhoidal serovars?

Line 98: Diarrheagenic escherichia coli (typing error). It should be made explicit that included enteropathogenic E. coli (EPEC), enterohemorrhagic (Shiga toxin-producing) E. coli (EHEC/STEC), enteroaggregative E. coli (EAEC), enterotoxigenic E. coli (ETEC), and enteroinvasive E. coli (EIEC). Most epidemiological reports from different countries are reported individually.

Author Response

Response: Thanks for your wonderful comments. Under your careful instructions, we have considered all of your precious suggestions in the revised manuscript and we have responded the questions you proposed point by point as follow.

1.The presentation of the data is very descriptive and sometimes repetitive in the discussion section. From the epidemiological standpoint, it would also be interesting to include the population of Zhejiang Province and a better description of the age group affected by the different foodborne diseases. Furthermore, it would be interesting to determine the incidence per 100,000 inhabitants within this surveillance.

Response: Thanks for your good suggestion. We strongly agree that it would be interesting to describe the age groups affected by different foodborne diseases and to determine the incidence per 100,000 inhabitants. However, data from the Foodborne Disease Outbreaks Surveillance System (FDOSS) are not suitable for such analysis. For outbreaks, information such as the age of the case is less important than information about the etiology and food that led to the outbreak. Our study also gives the rate of outbreak-related cases per million population, but these are only cases in outbreaks, and a large number of sporadic cases are not included. We believe that data from the Foodborne Disease Surveillance and Reporting System (FDSRS) are more suitable for the analysis you mentioned. In this system, we will collect the relevant information of each case, including gender, age, occupation, symptoms and signs, initial diagnosis, drug use and biological sample test results, and also collect information about suspicious food and dining places reported by the cases. Next, we will use data from FDSRS to do some research.

2.Table 2 could be shown as a figure.

Response: We have made the modification according to your suggestion.

3.Some information in the text could be summarized, reducing information already shown in the tables. Why is the data not compared with those of the European Union? European Unions have been reporting the incidence of foodborne illness for many years. See: https://www.ecdc.europa.eu/en/all-topics-z/food-and-waterborne-diseases-and-zoonoses/surveillance-and-disease-data/eu-one-health. Database: https://www.efsa.europa.eu/en/microstrategy/FBO-dashboard

Response: We have rewritten some sentences to make the text more concise. A comparison with the data of the European Union has also been added to the revised manuscript.

4.Many data with similar results appear in an article published by the same authors, see “Chen, L., Sun, L., Zhang, R., Liao, N., Qi, X., & Chen, J. (2022). Surveillance for foodborne disease outbreaks in Zhejiang Province, China, 2015–2020. BMC Public Health, 22(1), 2015–2020. https://doi.org/10.1186/S12889-022-12568-4”. Although the time is indeed limited, and it includes other non-biological hazards. However, part of the conclusions, as expected, are very similar.

Response: Thanks for your comments. The above article is an epidemiological analysis of all foodborne disease outbreaks reported in Zhejiang Province from 2015 to 2020. It describes all etiologies, including pathogenic bacteria, viruses, poisonous mushrooms, poisonous plants and their toxins, poisonous animals and their toxins, chemicals, etc. Through the above analysis, we found that pathogenic bacteria were the cause of the most outbreaks, so this manuscript provides a detailed analysis of this main cause over a longer period (2010-2020). Although pathogenic bacteria have been mentioned in previous articles, they are not as detailed as our current study.

5.Line 97: Salmonella enterica is non-typhoidal serovars?

 Response: Yes. Salmonella enterica is non-typhoidal serovars.

6.Line 98: Diarrheagenic escherichia coli (typing error). It should be made explicit that included enteropathogenic E. coli (EPEC), enterohemorrhagic (Shiga toxin-producing) E. coli (EHEC/STEC), enteroaggregative E. coli (EAEC), enterotoxigenic E. coli (ETEC), and enteroinvasive E. coli (EIEC). Most epidemiological reports from different countries are reported individually.

Response: Thanks for your helpful comments. In China's national food safety standard GB4789.6-2016, diarrheagenic Escherichia coli was defined as a kind of Escherichia coli that can cause diarrhea symptoms in humans and can cause human illness through contaminated food. Diarrheagenic Escherichia coli mainly includes EPEC, ETEC, EAEC, EIEC, etc. In our province, EAEC and EPEC are common causes of outbreaks.

Round 2

Reviewer 1 Report

Re Diarrheagenic Escherichia coli even if it is official in China, this term needs to be explained for a more general audience, such as Diarrheagenic Escherichia coli mainly includes EPEC, ETEC, EAEC, EIEC, and whatever other type. Since these are distinctive strains, most readers will be disappointed that the breakdown by strain is not shown (limitation). Make sure that the species name Burkholderia Gladioli has a lower case first letter in gladioli. Since Proteus vulgaris and Aeromonas are not typically pathogenic but can be common organisms present in food samples, I am concerned that the laboratory confirmation data may be suspect without further information. These need to be flagged for further inquiry or the authors need to indicate that these so-called pathogens may be treated with caution.

Author Response

Response: Thanks for your valuable comments. We have responded the questions you proposed point by point as follow.

1. Diarrheagenic Escherichia coli even if it is official in China, this term needs to be explained for a more general audience, such as Diarrheagenic Escherichia colimainly includes EPEC, ETEC, EAEC, EIEC, and whatever other type. Since these are distinctive strains, most readers will be disappointed that the breakdown by strain is not shown (limitation).

 Response: Thank you for your thoughtful advice. We have explained “diarrheagenic Escherichia coli” and added the limitations of the study in the discussion section, please see lines 93-95 and 235-237 of the revised manuscript.

2. Make sure that the species name Burkholderia Gladioli has a lower case first letter in gladioli.

Response: Thanks for your comment. All references to "Burkholderia Gladioli" in the text have been changed to "Burkholderia gladioli". Please refer to line 222, 223 and 227 of the revised manuscript.

3. Since Proteus vulgaris and Aeromonas are not typically pathogenic but can be common organisms present in food samples, I am concerned that the laboratory confirmation data may be suspect without further information. These need to be flagged for further inquiry or the authors need to indicate that these so-called pathogens may be treated with caution.

Response: Thanks for your good suggestion. We have added the description as you suggested, please see lines 96-97 of the revised manuscript.

Reviewer 3 Report

The manuscript has improved substantially after adequately responding to suggestions arising from the review process.

However, I would like to make some suggestions about minor mistakes:

                     Line 85: escherichia coli (Escherichia coli)

                     Figure 2 and 3: Revise italic name od microorganisms, and the name of Proteus vulgaris

                     Line 214: Burkholderia Gladioli (Burkholderia gladioli)

Author Response

Response: Thank you for your valuable comments and we are sorry for the mistakes. We have made changes one by one according to your suggestion.

1. Line 85: escherichia coli (Escherichia coli)

Response: We have corrected the mistake, please see line 87 of the revised manuscript.

 line 222, 223 and 227 of the revised manuscript.

2. Figure 2 and 3: Revise italic name od microorganisms, and the name of Proteus vulgaris

Response: We have redone Figures 2 and 3.

3. Line 214: Burkholderia Gladioli (Burkholderia gladioli)

Response: We have corrected the mistake, please refer to line 222, 223 and 227 of the revised manuscript.